# Isolation of Potential Probiotic *Bacillus* spp. from the Intestine of Nile Tilapia to Construct Recombinant Probiotic Expressing CC Chemokine and Its Effectiveness on Innate Immune Responses in Nile Tilapia

**DOI:** 10.3390/ani13060986

**Published:** 2023-03-08

**Authors:** Chatsirin Nakharuthai, Surintorn Boonanuntanasarn, Jirawadee Kaewda, Pimpisut Manassila

**Affiliations:** School of Animal Technology and Innovation, Institute of Agricultural Technology, Suranaree University of Technology, 111 University Avenue, Muang, Nakhon Ratchasima 30000, Thailand

**Keywords:** Nile tilapia, probiotics, CC chemokine, immune response, heterologous protein expression

## Abstract

**Simple Summary:**

The application of recombinant technology provides the possibility for the establishment of ideal probiotics as an alternative novel therapeutic approach by integrating promising heterologous proteins into the chromosome of conventional probiotics. In our previous study, we found that Nile tilapia CC chemokine genes had multiple roles in homeostatic functions in both immune and non-immune tissues. Moreover, they are crucially involved in the early immune responses to pathogens. Therefore, in this study, we isolated and characterized the potential probiotic *Bacillus subtilis* from the intestine of Nile tilapia to use as a live vehicle for delivering CC chemokine proteins directly to the intestine. Our results suggest that dietary supplementation with either wild-type or recombinant probiotics could enhance the immune responses of Nile tilapia, with stronger stimulation observed in the recombinant probiotics-supplemented group. It could be concluded that applying genetically engineered probiotics for developing novel strategies would provide greater health benefits through a combination efficacy of potential probiotics and desirable heterologous proteins.

**Abstract:**

This study aimed to investigate the potential probiotic *Bacillus* spp. from the intestine of Nile tilapia in order to construct a recombinant probiotic for the enhancement of the Nile tilapia immune response. One hundred bacterial isolates from the intestine of Nile tilapia were characterized for species identification using the 16s ribosomal RNA (rRNA). Only *Bacillus* isolates with exhibited antagonistic activity were investigated for their biological functions, which included protease-producing capacity, bile salts and pH tolerance, antibiotic susceptibility, and pathogenicity tests. According to the best results, *Bacillus* isolate B29, as closely related to *B. subtilis*, was selected to construct a recombinant probiotic for the delivery of CC chemokine protein (pBES*On*-CC). The existence of recombinant probiotics was confirmed by Western blotting before the feeding trial. In addition, the CC chemokine mRNA level was quantified in the intestine of fish fed probiotics after 30 days of feeding. Total immunoglobulin, lysozyme activity, alternative complement 50 activity (ACH50), and phagocytic activity of fish fed either wild-type or recombinant probiotics were significantly increased, indicating that probiotics could stimulate the Nile tilapia immune system through different processes. Interestingly, the dietary supplementation of recombinant probiotics has a stronger immune response enhancement than the wild-type strain.

## 1. Introduction

Nile tilapia (*Oreochromis niloticus*, Linn.) is Thailand’s most cultured freshwater fish, which produces an affordable source of high-quality proteins [1]. As such, the intensive culture of Nile tilapia in cage culture and the earthen pond in Thailand has been rapidly developed throughout the country to satisfy consumer demand. Although Nile tilapia demand has continuously increased, their supply has been limited due to the impact of climate change and the intensive system, which increases the risk of disease outbreaks and production losses. To control these problems, drug and chemical agents such as antibiotics have been extensively used. However, inappropriate antibiotic use exacerbates the development of antibiotic-resistant bacteria, which is a global health challenge in the aquaculture industry. In order to minimize drug usage, many researchers are attempting to develop alternative prophylactic and therapeutic strategies applicable to Nile tilapia aquaculture. One approach to minimize drug usage is the application of live biotherapeutic products such as probiotics for controlling disease outbreaks in various economic aquatic animals [2,3,4]. The interaction between probiotics and many types of host immune cells such as epithelial cells, monocytes, macrophages, dendritic cells, NK cells, B and T lymphocytes elucidate several desirable effects, which consist of improvement of epithelial and mucosal barrier function, competitive exclusion of pathogens by its binding ability to receptor sites of the host, production and secretion of extracellular enzymes and/or antimicrobial substances, maintenance of balanced gut microbiota, and modulation of both innate and adaptive immune responses [4,5,6,7]. In aquaculture systems, probiotics are now widely accepted as feed additives, feed supplementation, or added directly into the rearing water to improve the growth and health of fish as well as water quality [8,9,10,11,12,13]. A member of the genus *Bacillus*, including *B. subtilis*, *B. licheniformis, B. amyloliquefaciens*, *B. pumilus*, is considered to be among the important probiotics used in aquacultures that are present inside the fish body and the surrounding environment, such as rearing water and sediment [14,15]. Although many strains of *Bacillus* spp. have been successfully applied in the aquaculture industry, many health problems in aquatic animals still need to be addressed. The development of recombinant technology has led the way to produce heterologous proteins expressed in potential probiotics. Such recombinant probiotics can be used as a perspective vehicle to deliver the desired characteristics and functionalities of heterologous proteins directly to the intestine. Currently, the genus *Bacillus* is considered an acceptable biofactory to produce heterologous proteins for various purposes of basic research and industrial interest [16,17]. There are many advantages to employing *Bacillus* spp. such as the fact that they produce abundant quantities of products rapidly, are easily genetically modified for the expression and delivery of interested genes and are certified as generally recognized as safe (GRAS) by the US FDA [18,19,20]. 

Chemokine is one of the crucial immune genes that acts as a bridge linking the innate and adaptive immune systems. Their name is derived from their chemoattractive properties, which induce directed chemotaxis and recruit a variety of immune cells under both homeostatic and inflammatory conditions [21,22]. The chemokine family is divided into four major groups, CXC (α-chemokine), C (γ-chemokines), CX3C (δ-chemokines), and CC (β-chemokines), depending on their four characteristic cysteines and the spacing of their first two cysteine residues [23]. Of these, CC chemokine is the main group of chemokines described, which have been intensively cloned and characterized in several fish species, including rainbow trout (*Oncorhynchus mykiss*) [24,25], channel catfish (*Ictalurus punctatus*) [26], zebrafish (*Danio rerio*) [27], large yellow croaker (*Pseudosciaena crocea*) [28], rock bream (*Oplegnathus fasciatus*) [29], snakehead murrel (*Channa striatus*) [30], Nile tilapia (*Oreochromis niloticus*) [31], and olive flounder (*Paralichthys olivaceus*) [32]. In our previous study, we successfully cloned and characterized CC chemokine genes of Nile tilapia. The result strongly indicated that these genes are crucially involved in normal and pathological processes. Moreover, functional analyses clearly demonstrated that recombinant CC chemokine proteins efficiently enhanced the phagocytic activity (in vitro) to protect Nile tilapia from pathogens [31].

Therefore, the aim of this study was to isolate and characterize the potential probiotic *Bacillus* spp. from Nile tilapia’s intestine through their biological functions, including antagonistic activity, protease-producing capacity, bile salts and pH tolerance, antibiotic-susceptibility, and pathogenicity tests. According to these results, the potential probiotic was selected to construct a recombinant probiotic for the delivery of CC chemokine protein (pBES*On*-CC). Western blot analysis was employed to confirm the expression of CC chemokine in potential *Bacillus* spp. before the feeding trial. At the end of the feeding trial (30 days), the effects of dietary supplementation with the wild-type strain and the recombinant probiotic on innate immune responses were investigated. In addition, real-time quantitative PCR was used to quantify the CC chemokine mRNA expression in the intestine of the experimental fish. In this sense, the modern method of genetically engineered probiotics expressing immune-related genes may provide new strategies for simultaneously enhancing immunity and microbiota modulation.

## 2. Materials and Methods

### 2.1. Isolation of Potential Probiotic Bacillus spp.

#### 2.1.1. Experimental Animals

Nile tilapia (~100 g) were acclimatized in the earthen pond with an aeration system at the Suranaree University of Technology Farm in Nakhon Ratchasima province, Thailand. Fish were fed ad libitum twice a day with a 10.0 g kg^−1^ Jerusalem artichoke (*Helianthus tuberosus*) supplemented diet for 45 days. The described experiment was conducted according to the Ethics Committee of Suranaree University of Technology, Animal Care and Use Committee (approval no. SUT-IACUC-003/2022). 

#### 2.1.2. Bacterial Isolation and Characterization

Bacterial species were isolated from the intestine of ten fish in Section 2.1.1. The anterior, middle, and posterior portions of the intestine were excised at approximately 1 cm and washed with 0.85% (*w*/*v*) sodium chloride solution. Consequently, the samples were minced with sterile scissors and homogenized with a sterile pestle. After that, all samples were serially diluted 10-fold, and each dilution was cultured on Tryptic Soy Agar (TSA) (HiMedia Laboratories Pvt., Maharashtra, India), followed by incubation for 16–18 h at 30 °C. The single colony was subjected to Gram’s staining to determine the bacterial morphology and to classify Gram-positive and negative bacteria using a compound microscope. To allow the formation of *Bacillus* spore and eliminate other contaminating bacteria, Gram-positive bacteria were heated at 80 °C for 20 min, quickly chilled on ice for 1–2 min, and cultured on TSA, followed by incubation for 16–18 h at 30 °C. The single colonies were preserved with 20% glycerol at −80 °C for further analysis. The isolated bacteria were characterized for species identification by colony PCR analysis using the 16s ribosomal RNA (rRNA) universal primers 27f (5′-AGAGTTTGATCMTGGCTCAG-3′) and 1492r (5′-TACGGYTACCTTGTTACGACTT-3′) [33]. The PCR was performed using GoTaq^®^ DNA Polymerase (Promega Corporation, Madison, WI, USA) in a total volume of 30 µL, which contained 0.2 mM of each dNTP, 10 pmol of each primer, 1.5 mM MgCl_2_, 5× buffer GoTaq^®^ Reaction Buffer, and 1 U GoTaq^®^ DNA Polymerase. For the PCR reaction, initial denaturation was conducted at 95 °C for 5 min followed by 35 reaction cycles, each consisting of a denaturation step at 95 °C for 30 s, annealing at 55 °C for 30 s, and extension at 72 °C for 45 s, with a final extension step at 72 °C for 5 min. The expected amplicon sizes were purified using a FavorPrep GEL/PCR Purification Kit according to the manufacturer’s recommended protocol. Subsequently, the purified PCR was sequenced by the Macrogen sequencing service (Macrogen Inc., Seoul, Republic of Korea) using the 16s rRNA universal primers 27f and 1492r. After sequencing, the resulting nucleotide sequences were searched for bacterial species identification in the GenBank database (http://www.ncbi.nlm.nih.gov) using BLASTN program.

#### 2.1.3. Antagonistic Activity against Pathogenic Bacteria

To screen probiotic bacteria with antagonistic activity, the cross-streak method was performed, as described by Sookchaiyaporn et al. [34], with slight modification. Briefly, the isolated bacteria from Section 2.1.2 and bacterial pathogenic strains, including *Streptococcus iniae*, *Streptococcus agalactiae,* and *Aeromonas hydrophila,* were grown in Tryptic Soy Broth (TSB) (HiMedia Laboratories Pvt., Maharashtra, India) for 16–18 h at 30 °C. After incubation, the concentration of isolated bacteria was adjusted to 1 × 10^6^ CFU/mL and 1 × 10^8^ CFU/mL, while pathogenic bacteria were adjusted to 1 × 10^8^ CFU/mL, respectively. A loopful of 1 × 10^6^ and 1 × 10^8^ of each isolated bacterium was streaked onto TSA as the first and second line, followed by the perpendicular streaking of 1 × 10^8^ CFU/mL bacterial pathogenic to isolated bacteria, then incubated for 72 h at 30 °C. The bacterial antagonistic activity was determined by observing the inhibition zone size at 24 h, 48 h, and 72 h, respectively. Each isolated strain was performed in triplicates.

#### 2.1.4. Phylogenetic Analysis

According to the antagonistic activity result, the potential probiotic *Bacillus* spp. was used to construct the phylogenetic tree with 1000 bootstrap replications by the MEGA X program using the maximum-likelihood phylogeny method with MUSCLE alignment [35,36].

#### 2.1.5. Determination of Bile Salts Tolerance

To determine the bile salts tolerance, the isolated bacteria were inoculated in TSB enriched with 0.5, 1, and 2% bile salts and incubated at 30 °C for 1, 2, 4, and 6 h, respectively. During the 1, 2, 4, and 6 h incubation, the bacterial viability was carried out by streaking on TSA followed by incubation overnight at 30 °C. Each isolated strain was performed in triplicates.

#### 2.1.6. Determination of pH Resistance

To determine pH resistance, the isolated bacteria were inoculated in TSB with a varying pH of 2, 3, 4, 5, 6, 7, 8, and 9 using HCl or NaOH and incubated at 30 °C for 1, 2, 4, and 6 h, respectively. During the 1, 2, 4, and 6 h incubation, the bacterial viability was carried out by streaking on TSA followed by incubation overnight at 30 °C. Each isolated strain was performed in triplicates.

#### 2.1.7. Antibiotic Susceptibility

The antibiotic resistance was conducted by the agar disk-diffusion method [37]. Ten types of antibiotics, including ampicillin, neomycin, oxytetracycline, ciprofloxacin, enrofloxacin, erythromycin, sulphamethoxazole, tetracycline, amoxycillin, and nalidixic acid, were used in the assay. Briefly, The McFarland standards were used as a reference to adjust the turbidity of bacterial suspensions to 10^6^ CFU/mL. The Mueller Hinton agar plates were inoculated with tested bacteria, and then antibiotic discs were placed on the agar surface, followed by incubation overnight at 30 °C. The diameters of inhibition growth zones were measured in millimeters. The results were classified as susceptible, intermediate, or resistant according to the Clinical and Laboratory Standards Institute (CLSI) criteria [38]. Each isolated strain was performed in triplicates.

#### 2.1.8. Determination of Protease-Producing Capacity

To determine the proteolytic activity, the isolated bacteria were grown in TSB, and the concentration was adjusted to 1 × 10^8^ CFU/mL. The sterile filter paper was placed onto the surface of TSA containing 5% skim milk (Merck KGaA, Darmstadt, Germany). Then, 10 µL of bacterial suspension was spotted onto sterile filter paper, followed by incubation at 30 °C for 24 h. After incubation, the protease-producing capacity was determined by the presence of a clear hydrolysis zone surrounding the filter paper and was evaluated by measuring the diameter of the clear zone. Each isolated strain was performed in triplicates.

#### 2.1.9. Safety and Pathogenicity Assessments

To evaluate the safety of probiotic candidates, twenty healthy Nile tilapia (100 ± 5 g) were randomly divided into four tanks (5 fish/tank). The experimental fish were acclimatized in 700-L fiberglass tanks containing clean and fully oxygenated fresh water for 7 days prior to the beginning of the experiment. After acclimatization, the fish were intraperitoneally injected with 0.1 mL of 0.85% NaCl as a control group and 1 × 10^4^, 1 × 10^8^, 1 × 10^12^ CFU/mL of isolated bacteria for the evaluation of the lethal dose 50 (LD50) toxicity. Then, physiological changes and cumulative mortality were observed every day for 15 days.

### 2.2. Construction of Recombinant Probiotic Bacillus Isolate B29 Expressing CC Chemokine

To construct recombinant *Bacillus* isolate B29 expressing CC chemokine (pBES*On*-CC), the forward primer containing *Bam*HI and *Hind*III restriction site and the reverse primer followed by *Xho*I and *Hind*III restriction site (Table 1) were designed from cDNA encoding *On*-CC1 chemokine proteins as our previously described [31] with the following conditions: initial denaturation was conducted at 95 °C for 3 min; 35 reaction cycles of a denaturation step at 95 °C for 30 s, annealing at 55 °C for 30 s, and extension at 72 °C for 45 s; and a final extension step at 72 °C for 5 min. The obtained PCR products were purified using FavorPrep GEL/ PCR Purification Kit according to the manufacturer’s recommended protocol, ligated into the pGEM^®^T-Easy plasmid (Promega Corporation, Madison, WI, USA), and transformed into *E. coli* (DH5α) competent cells. After that, plasmids were extracted, and then 1 µg of extracted plasmids were double digested with restriction enzyme (*Bam*HI and *Hind*III) to verify the insertion of the correct DNA fragment. To confirm correct in-frame insertion, the plasmid DNA, which appeared to contain inserted DNA, was sequenced by the Macrogen sequencing service (Macrogen Inc., Seoul, Republic of Korea) using the sp6 and T7 promoter primers. Subsequently, the recombinant plasmid was further double-digested with *Bam*HI and *Hind*III restriction enzymes. The digested DNA fragments were purified and ligated into a linearized pBES expression vector (Takara Bio USA, Inc., San Jose, CA, USA) double-digested with *Bam*HI and *Hind*III restriction enzymes, respectively. Then, the ligated DNAs were transformed into *E. coli* (DH5α) competent cells, and the positive colonies were examined by PCR prior to plasmid DNA extraction. *Bacillus* isolate B29 from Section 2.1.2 was used to prepare competent cells according to the method of Xue et al. [39], with slight modifications. The positive plasmid was transformed into *Bacillus* isolate B29 competent cells for protein expression. The resulting transformants were selected on Luria-Bertani (LB) agar plates containing 100 µg/mL of kanamycin. Finally, the positive clones were proved by double restriction enzyme digestion followed by agarose gel electrophoresis and were sequenced by the Macrogen sequencing service (Macrogen Inc., Seoul, Republic of Korea) using the specific primers (H-B-*On*CCF and H-X-*On*CCR).

### 2.3. Western Blot Analysis

*Bacillus* isolate B29 strain integrated with recombinant CC chemokine protein was grown in LB broth containing 100 µg/mL of kanamycin at 37 °C with 180 rpm agitation for 18 h. Bacterial cell cultures were harvested by centrifugation at 5000 rpm for 10 min. The pellet was resuspended in 1 × phosphate-buffered saline (PBS, pH 7.4) and sonicated for 3 min in an ice bucket, with 2 s each. Western blot analyses were applied to confirm the existence of secreted Nile tilapia CC chemokine produced from *Bacillus* isolate B29. The recombinant CC chemokine protein was separated on SDS-PAGE gel and transferred onto a nitrocellulose membrane using a Trans-Blot^®^ Turbo^TM^ (Bio-Rad Laboratories, Hercules, CA, USA)) at 25 V, 1.0 A for 30 min. To minimize non-specific binding, the membrane was blocked with 2% skim milk at room temperature for 1 h. The Western blot assays were performed using anti-His tag mouse monoclonal antibody and goat anti-mouse IgG with horseradish peroxidase (HRP). The target proteins were detected using SuperKine™ West Pico PLUS Chemiluminescent Substrate (Abbkine Scientific Co., Ltd., Redlands, CA, USA) and visualized with enhanced chemiluminescence (ECL) solution using ChemiDoc MP Imaging System^TM^ (Bio-Rad Laboratories, Hercules, CA, USA).

### 2.4. Effects of Dietary Probiotic Supplementation on Gene Expression and Innate Immune Response

#### 2.4.1. Experimental Design

Sixty Nile tilapia weighing 99.37 ± 5.57 g were maintained at the Suranaree University of Technology Farm (SUT Farm; Nakhon Ratchasima, Thailand). The experimental design was completely randomized with three groups, where 20 fish for each group were acclimatized into the 3 cement ponds (2 × 2 × 1 m^3^) containing clean freshwater with an aeration system for two weeks. After acclimatization, group 1 was fed a commercial diet + 0.85% NaCl as a control group, group 2 was fed a commercial diet + wild-type *Bacillus* isolate B29, and group 3 was fed a commercial + recombinant *Bacillus* isolate B29 expressing CC chemokine. Throughout the experimental period, all groups were fed ad libitum twice daily for 30 days.

#### 2.4.2. Diet Preparation

Prior to this experiment, wild-type *Bacillus* isolate B29 and recombinant *Bacillus* isolate B29 from Section 2.1.2 and Section 2.2 were aliquoted and stored in glycerol stocks at −80 °C. For each experimental diet preparation, an aliquot of glycerol-stock bacteria was streaked onto a TSA plate and incubated at 37 °C for 18–24 h. After that, the starter culture was prepared by inoculating 5 mL TSB with a single colony of each bacterium (wild-type *Bacillus* isolate B29 and recombinant *Bacillus* isolate B29) and incubated at 37 °C for 18–24 h, with a shaking rate of 180 rpm. After cultivation, the starter culture was then transferred into a new flask containing 500 mL TSB and incubated in the incubator shaker with a shaking rate of 180 rpm at 37 °C for 18–24 h. Each bacterial suspension was harvested by centrifugation at 3500 rpm for 15 min. The bacterial pellet was washed twice with sterile 0.85% NaCl solution and resuspended in sterile 0.85% NaCl. The concentrations of each bacterium were adjusted to 1 × 10^8^ CFU/mL in a total volume of 200 mL. After that, each bacterial suspension was mixed with 1 kg of a commercial diet to obtain a final probiotic concentration of 1 × 10^8^ CFU/kg. For the control group, 1 kg of a commercial diet was mixed with 200 mL of sterile 0.85% NaCl. To remove excess moisture, all experimental diets were air-dried at room temperature and kept in the refrigerator until use. All experimental diets were prepared weekly.

#### 2.4.3. Fish Sampling and Collection

After the first feeding trial, three fish from each group were randomly sampled at 6, 24, 48 h, 7, 15, and 30 days, respectively. Before sampling, fish were anesthetized with a concentration of 0.3 mL/L 2-phenoxyethanol. The intestine samples were collected from each fish for qRT-PCR analysis, while serum samples were collected for immune parameters evaluation, including lysozyme activity, total immunoglobulin (Ig), and alternative complement (ACH50) activity.

#### 2.4.4. Expression Level Analysis of CC Chemokine mRNA of Nile Tilapia after Feeding with Experimental Diets Using Real-Time Quantitative PCR

##### Gene Cloning

Cloning of a partial gene fragment coding for CC chemokine and β-actin was performed to evaluate the expression level of CC chemokine mRNAs of Nile tilapia after feeding with experimental diets. Total RNA was extracted from 100 mg of spleen using Trizol^®^ reagent (Invitrogen Corporation, Carlsbad, CA, USA) and RNase-free DNase I (Promega Corporation, Madison, WI, USA) according to the manufacturer’s instructions. The first-strand cDNA was subsequently synthesized using the ImProm-II™ Reverse Transcription System kit (Promega Corporation, Madison, WI, USA). Primers for the amplification used in this study are shown in Table 1. In a total volume of 40 µL, each PCR reaction was conducted with 200 µM of each dNTP, 1 µM of each primer, 2.5 mM MgCl_2_, 1.0 × buffer Ex Taq^TM^, and 1.25 U Ex Taq^TM^. The cycling reactions were carried out at 95 °C for 3 min for an initial denaturation followed by 40 cycles, each consisting of at 95 °C for 30 s, at 55 °C for 30 s, and at 72 °C for 30 s, and the final elongation step at 72 °C for 5 min. The PCR product of the expected size was purified using a FavorPrep GEL/PCR Purification Kit according to the manufacturer’s instructions. The purified DNA was cloned into pGEM^®^ T-Easy plasmid (Promega Corporation, Madison, WI, USA). The plasmids were sequenced by Macrogen, Inc. (Seoul, Republic of Korea) and stored for further use as a standard for quantitative RT-PCR (qRT-PCR).

##### Total RNA Extraction and qRT-PCR Analysis

To evaluate the expression level of CC chemokine mRNAs of Nile tilapia after the feeding trial, the intestine was collected from fish in each replication trial. Total RNA was extracted using Trizol reagent and Rnase-free DNase I (Promega Corporation, Madison, WI, USA) according to the manufacturer’s recommended protocol. First-strand cDNA was synthesized using the ImProm-II^TM^ Reverse Transcription System kit (Promega Corporation, Madison, WI, USA) and subjected to qRT-PCR analysis (in triplicate) using LightCycler^®^ 480 SYBR Green I Master Mix (Roche Applied Science, Indianapolis, IN, USA) according to the manufacturer’s recommended protocol. The primers used in this experiment were designed following our previous study [31]. The CC chemokine gene expression level was normalized to the expression of β-actin as an internal reference. PCR samples were prepared in a final volume of 10 µL consisting of 5 µL of LightCycler^®^ 480 SYBR Green I Master Mix, 1 µL of forward primer, 1 µL of reverse primer, 2 µL of ddH_2_O, and 1 µL of cDNA template. The PCR conditions were 95 °C for 5 min followed by 40 reaction cycles, each consisting of a denaturation step at 95 °C for 15 s, annealing at 55 °C for 15 s, and extension at 72 °C for 15 s, with a final elongation step at 72 °C for 5 min. DNA melting curve analysis was performed to verify the specificity of the primers. The mRNA level of CC chemokine was quantitatively analyzed, as described in Boonanuntanasarn et al. [40].

#### 2.4.5. Immune Assays

##### Lysozyme Activity

Lysozyme activity was measured according to the method of Siwicki et al. [41]. Briefly, lyophilized hen egg whites at concentrations of 0, 2.5, 5, 10, 15, and 20 µg/mL in 0.06 M PBS (pH 6.0) were prepared as reference standards for the standard curve. After that, 10 µL of Nile tilapia serum and each concentration of reference standard were added into wells of a 96-well flat bottom plate (in triplicate). To determine serum lysozyme activity, a suspension of 190 µL of 0.2 mg/mL dried *Micrococcus lysodeikticus* (ATCC 4698; Sigma-Aldrich, St. Louis, MO, USA) was quickly added into all wells of the plate. The reaction was carried out at 25 °C, and the optical density (OD) was measured at 450 nm at the initial time (0 min) and 60 min using a microplate spectrophotometer (BioTek™ EPOCH, Agilent Technologies, Santa Clara, CA, USA). The concentrations of lysozyme (µg/mL) were calculated from a standard curve of known hen egg white concentrations.

##### Total Immunoglobulin (Ig)

The total Ig concentration (mg/mL) in Nile tilapia serum was determined using the total protein kit (Biuret method; Erba, Mannheim, Germany) according to the method of Siwicki et al. [42]. Briefly, Nile tilapia serum (10 µL) was added in a 1.5 mL microcentrifuge tube, followed by the addition of an equal volume of 12% PEG solution (polyethylene glycol; Sigma-Aldrich, St. Louis, MO, USA). The mixing samples were incubated at 25 °C for 30 min and centrifuged at 3000× *g* for 5 min at 4 °C. Following centrifugation, the mixture was separated into supernatant (non-Igs) and pellet (total Igs). After that, 4 µL of the supernatant, serum (total protein), and standard bovine serum albumin (Sigma-Aldrich, St. Louis, MO, USA) were added into a 96-well flat bottom plate in triplicate. The protein concentration was determined according to the manufacturer’s recommended protocol, and the OD was measured at 546 nm using a microplate spectrophotometer (BioTek™ EPOCH, Santa Clara, CA, USA). After precipitation, total Igs were calculated by subtracting total protein from non-Igs. The concentrations of proteins were derived from the standard curve constructed with bovine serum albumin.

##### Alternative Complement (ACH50) Activity

ACH50 activity was determined following a previously described method [43]. Briefly, Nile tilapia serum (50 µL) was two-fold serial diluted with EGTA-GVB buffer (gelatin veronal buffered saline, 10 mM ethyleneglycol-bis (beta-amino-ethyl ether) *N*-*N*’-tetraacetate) and then an equal volume of 5 × 10^7^ cells/mL of GRBC suspension was added to the diluted plasma. After that, the mixture was incubated for 90 min at 25 °C and centrifuged at 3000× *g* for 10 min at 4 °C. The OD was measured at 415 nm using a microplate spectrophotometer (BioTek™ EPOCH, Santa Clara, CA, USA). Nile tilapia diluted serum representing the volume of complement producing 50% hemolysis (ACH50) of GRBCs was determined, and the number of ACH50 units/mL was calculated for each experimental group (in triplicate).

##### Phagocytic Activity Analysis

The percentage of phagocytic activity (PA) and the phagocytic index (PI) were determined by appropriately modifying the methods described by Puangkaew et al. [44]. Peripheral blood leukocytes (PBLs) were used in this experiment. Briefly, one mL of whole blood was withdrawn from the caudal vein of the fish using a sterile syringe coated with an anticoagulant (K_2_EDTA) solution. The unclotted blood was transferred to the 15 mL conical tube containing 2 mL of RPMI medium and put into the 15 mL conical tube containing Histopaque^®^-1077 (Sigma-Aldrich, St. Louis, MO, USA). The two-layer mixture was then centrifuged at 400× *g* for 30 min at 25 °C in a swing rotor centrifuge, 3 mL of the opaque interface was gently transferred into a new 15 mL, and an equal volume of phosphate-buffered saline PBS (pH 7.4) was added. The sample was mixed gently and then centrifuged 2 times at 250× *g* for 10 min. The blood pellet was dissolved in PBS (pH 7.4), and the phagocyte concentration was adjusted to 1 × 10^6^ cells/mL. The sample from each experimental group was loaded onto the 22 × 22 mm coverslip and allowed for adhesion to the coverslip’s surface for 2 h. All of these experimental groups were performed in triplicate. The adherent phagocytic cells were incubated for 1.5 h at room temperature with 1 × 10^7^ particles of latex beads (Sigma-Aldrich, St. Louis, MO, USA) prepared in 200 µL PBS (pH 7.4). The unattached cells and excess beads were then washed three times with PBS (pH 7.4). The attached cells and beads were stained with Diff-Quick staining dye (Fisher Scientific, Waltham, MA, USA). The attached cells and beads were stained with Diff-Quick staining dye (Fisher Scientific, Waltham, MA, USA). At least 300 phagocytosing and non-phagocytosing cells and the number of beads in/on the phagocytizing cells were observed under 100× light microscopy. Finally, the percentage of phagocytic activity (PA) and the phagocytic index (PI) were determined using the equations below. PA and PI statistical analysis was performed as previously reported [31].
PA = (number of phagocytic cells/number of total cells count) × 100
PI = (number of total bead particles/number of total phagocytic cells) × 100

### 2.5. Statistical Analysis

All statistical analyses were conducted using SPSS software ver.20 (SPSS Inc., Chicago, IL, USA). Statistical differences between groups were carried out using a one-way analysis of variance, followed by the post hoc Tukey’s test to assess the significance of differences between the groups. The effects and differences were determined significant throughout the experiment at a probability value that was less than 0.05 (*p* < 0.05).

## 3. Results

### 3.1. Bacterial Isolation and Characterization

A total of one hundred bacterial strains were isolated from the intestinal tract of healthy Nile tilapia after feeding with Jerusalem artichoke (*Helianthus tuberosus*) for 45 days. Of these, 36 bacterial isolates were gram-positive with a bacilli shape and catalase positive, and only 13 were *Bacillus* spp. after identifying bacterial species through heat and cold shock methods followed by colony PCR analysis using the 16S ribosomal RNA (rRNA). According to the result, 16S rRNA sequence analysis confirmed that bacterial isolates displayed approximately a 75.65–99.24% similar identity to other known *Bacillus* spp. (Table 2).

### 3.2. Antagonistic Activity against Pathogenic Bacteria

To evaluate the possibility of antibacterial activity against pathogens in Nile tilapia, an antagonistic assay was used as the first criterion for selecting an effective probiotic. The result showed that only four *Bacillus* isolates (B28, B29, B30, and B31) displayed inhibitory and/or colonization activities against pathogenic bacteria in Nile tilapia, including *A. hydrophila*, *S. agalactiae*, and *S. iniae*. Interestingly, there were only 1 × 10^6^ CFU/mL and 1 × 10^8^ CFU/mL of *Bacillus* isolate B29 that could exhibit antibacterial activity against all three pathogenic bacteria (colonization and/or inhibition), as shown in Figure 1.

### 3.3. Phylogenetic Analysis

To confirm the evolutionary relationship of selected *Bacillus* isolates (B28, B29, B30, and B31), a phylogenetic tree analysis was performed by comparing their 16s rRNA to that of other known *Bacillus* species from the GenBank database. The result confirmed that all four isolates, which exhibited antagonistic activity, were clustered within *Bacillus* spp. In addition, *Bacillus* isolates B28, B30, and B31 were closely related to *B. pumilus*, whereas *Bacillus* isolate B29 was closely related to *B. subtilis* (Figure 2). *Streptococcus agalactiae* (NR_040821.1) was used as the outgroup in the phylogenetic tree.

### 3.4. pH and Bile Salts Tolerance Test

In vitro pH and bile salt tolerance tests showed that all four isolates (B28, B29, B30, and B31) could survive under simulated gastrointestinal tract conditions. The result demonstrated that they could grow in LB broth at pH 2–9 for 6 h and LB containing 0.5–2% bile salts for 6 h, as shown in Figure 3. However, the viability of *Bacillus* isolate B28 dramatically declined in LB containing 0.5, 1, and 2% of bile salts at 4 and 6 h.

### 3.5. Protease-Producing Capacity

The protease-producing capacity of isolated bacteria was evaluated by measuring the diameter of a clear zone surrounding the bacterial colonies. The result showed that three isolates, including *Bacillus* isolate B29, B30, and B31, could produce a proteolysis zone surrounding the colonies by proteolytic action, as shown in Table 3. Interestingly, the best result is shown in *Bacillus* isolate B29 (Figure 4)

### 3.6. Antimicrobial Susceptibility Profile

Based on previous results, including antagonistic activity, pH and bile salts tolerance, and protease-producing capacity, only *Bacillus* isolate B29 was selected to determine the susceptibility test with ten commercial antibiotics. The result showed that *Bacillus* isolate B29 was susceptible to all tested antibiotics, including ampicillin (10 µg), neomycin (30 µg), oxytetracycline (30 µg), ciprofloxacin (5 µg), enrofloxacin (5 µg), erythromycin (15 µg), sulfamethoxazole (15 µg), tetracycline (30 µg), amoxycillin (10 µg), and nalidixic acid (30 µg) (Figure 5).

### 3.7. Safety of Candidate Strains to Nile Tilapia

To assess the safety of *Bacillus* isolate B29, this bacterium’s lethal dose 50 (LD50) was performed. The result showed that LD50 could not be detected in this study. *Bacillus* isolate B29 did not reveal any physiological changes and cumulative mortality in experimental fish, even though the highest dose of injection at 1 × 10^12^ CFU/mL was used in this experiment. Therefore, this bacterium is considered possibly safe with no observed adverse effect level.

### 3.8. Production of Recombinant Probiotic Expressing CC Chemokine in Nile Tilapia and Western Blot Analysis

The recombinant *Bacillus* isolate B29 expressing CC chemokine (pBES*On*CC) was constructed by cloning the Nile tiapia CC chemokine gene (*On*-CC1) into the shuttle vector pBES. The positive clone confirmed whether the plasmid was correctly transformed into *Bacillus* isolate B29 using restriction enzymes (*Bam*HI, *Hind*III) digestion and sequencing (Figure 6a). The presence of secreted Nile tilapia CC chemokine produced from *Bacillus* isolate B29 was confirmed by Western blot analysis. As shown in Figure 6b, the molecular weight of CC chemokine protein was approximately 16 kDa, consistent with our previous study [31].

### 3.9. Expression Level Analysis of CC Chemokine mRNA of Nile Tilapia after Feeding with Experimental Diets Using Real-Time Quantitative PCR

In this study, real-time quantitative PCR was used to determine the expression level of the CC chemokine mRNA of Nile tilapia supplemented with recombinant *Bacillus* isolate B29 expressing CC chemokine after 30 days of feeding. The expression level of CC chemokine mRNA was significantly increased in the intestine of Nile tilapia fed a recombinant probiotic-supplemented diet, as shown in Table 4.

### 3.10. Effects of Wild-Type and Recombinant Probiotic Supplemented Feed on Innate Immune Responses of Nile Tilapia

As shown in Figure 7, the result revealed that fish fed with a wild-type and recombinant *Bacillus* isolate B29 expressing CC chemokine supplemented diet for the experimental period of 30 days led to an enhance immune response, including ACH50, lysozyme activity, and total immunoglobulin compared to the control (*p* < 0.05). Significant increases in ACH50 were observed at 24 h, 48 h, and 7 days after feeding with experimental diets. Although ACH50 levels at 6 h, 15 days, and 30 days were not significantly different compared to the control, they were also higher in probiotic supplementation groups with a high standard deviation. Total Ig of fish fed with recombinant probiotic-supplemented diet was significantly increased at 6 h, 7, 15, and 30 days, while wild-type strain supplementation groups were significantly higher than the control group at 7, 15, and 30 days, respectively. In addition, significant increases in lysozyme activity were observed at 15 and 30 days after feeding both the wild-type and recombinant probiotic groups.

After 30 days of feeding, phagocytic activity results showed that both wild-type and recombinant probiotic-supplemented groups significantly increased the PA of the Nile tilapia phagocytes by 31.40 ± 3.44 and 41.21 ± 12.28%, respectively, compared to the control (19.80 ± 1.01) (Figure 8a). Interestingly, fish fed with 1 × 10^8^ CFU/g recombinant probiotic exhibited significantly increased PI, with values of 2.13 ± 0.18, compared to the control (1.37 ± 0.07) and wild-type strain-supplemented group (1.55 ± 0.06) (Figure 8b).

## 4. Discussion

Nowadays, the application of recombinant technology provides the possibility for the establishment of ideal probiotics as an alternative novel therapeutic approach by integrating promising heterologous proteins into the chromosome of conventional probiotics. Gram-positive probiotic bacteria, especially the genus *Bacillus*, an autochthonous bacteria, is one of the most extensively studied and used as a promising host for various industrial heterologous protein production [45,46]. In Nile tilapia farming, *Bacillus* spp. is considered a crucial probiotic for developing practical strategies to protect Nile tilapia from pathogenic infection. Until now, novel probiotic *Bacillus* spp. have been identified and investigated their ability in several aquatic animal species, such as striped catfish (*Pangasianodon hypophthalmus*) [47], rohu (*Labeo rohita*) [48,49], Pla-mong (*Pangasius bocourti*) [50], Asian sea bass (*Lates calcarifer*) [51], Mozambique tilapia (*Oreochromis mossambicus*) [52], Guppy (*Poecilia reticulata*) [53], olive flounder (*Paralichthys olivaceus*) [54], cobia (*Rachycentron canadum*) [55], white shrimp (*Litopenaeus vannamei*) [56]. In this study, we isolate and characterize potential probiotic *Bacillus* spp. from the intestine of Nile tilapia. Thirteen out of one hundred isolates were identified as a member of *Bacillus* spp. using both conventional methods (e.g., Gram staining, microscopic morphology, endospore formation, catalase test, and a simple heat/cold shock method) and molecular techniques. Molecular identification was performed to identify bacteria at the species level. The antagonistic activity was used as the first criterion for evaluating potential probiotics because it indicates their adhesion and aggregation ability to the gut mucosa and the production of antimicrobial compounds [57,58]. This study has demonstrated that four out of thirteen *Bacillus* isolates displayed antagonistic activities against pathogenic bacteria in Nile tilapia, including *A. hydrophila*, *S. iniae*, and *S. agalactiae,* at different levels. Generally, *Bacillus* spp. is also known as a biocontrol agent that plays a vital role in inhibiting the growth of a wide range of pathogens due to the production of antimicrobial metabolites, especially antimicrobial compounds (AMCs) (e.g., bacteriocins, bacteriocin-like inhibitory substances, siderophores, lysozymes, proteases, chitinase, lipopeptide), quorum-quenching enzymes, and other antibacterial metabolic products [59,60]. The antimicrobial ability against pathogens reinforces the *Bacillus* strain as an attractive alternative to antibiotics for combating disease problems in the aquaculture industry. In our study, *Bacillus* isolate B29 was a promising probiotic that exerted the highest interventions to inhibit and/or colonize pathogenic bacteria in Nile tilapia. This finding was consistent with those of previous studies describing antagonistic properties of *Bacillus* spp., which is isolated from the gastrointestinal tract of many aquatic animal species such as *Litopenaeus vannamei* [61], *Penaeus monodo* [62], *Paralichthys lethostigma* [63] and *Oreochromis niloticus* [34]. Among probiotic *B. subtilis* strains, *Bacillus* isolate B29 in this study, with remarkable antagonistic activity against three important pathogenic bacteria in Nile tilapia, could be a notable candidate *Bacillus* spp. with applications in the intensive culture system of Nile tilapia farming to reduce drug and chemical application. However, the antagonistic mechanism of *Bacillus* spp. against pathogens has yet to be thoroughly documented; therefore, further study about this mechanism is necessary to provide helpful information for better understanding and developing effective strategies.

To confirm the authenticity of isolated bacterial strains, a phylogenetic tree of 16S rRNA genes from *Bacillus* spp. was constructed using nucleotide sequences of four *Bacillus* isolates in our study and other known *Bacillus* sequences available in GenBank database. The result demonstrated that *Bacillus* isolate B28, B30, and B31 were closely related to *B. pumilus*, whereas *Bacillus* isolate B29 was related to *B. subtilis*, consistent with the 16S rRNA molecular analysis. Consequently, the selection criteria for probiotic strains, which consisted of pH and bile salts tolerance and protease-producing capacity, were further used to assess their biological activities. According to the in vitro bile salt and pH tolerance test, *Bacillus* isolate B29 has substantial tolerance to bile salt and pH compared with other isolates. In general, the pH value of the digestive tract of Nile tilapia ranges from 2.5–8, and the physiological concentration of bile in Nile tilapia’s digestive tract was from 0.4 to 1.3% [64]. This finding confirmed that *Bacillus* isolated B29 with exhibited strong antagonistic activity could be able to survive and colonize throughout the gastrointestinal tract of Nile tilapia. It is possible to compete for the exclusion of pathogenic bacteria in Nile tilapia since the colonization of probiotics in the digestive tract environment could protect the fish from the establishment of pathogenic bacteria. In addition, several studies have also reported that *Bacillus* spp. can develop biofilms to increase their tolerance to acid and bile in the GI tract of fish [65,66]. Moreover, the suppression of pathogenic bacteria caused by antagonistic effects not only enhanced a proper balance of the intestinal microbial flora but also reduced antigenic stimulation resulting in the reduction of the immune cells in the gastrointestinal mucosa. This event may contribute to improved intestinal absorption and utilization of nutrients [67,68,69,70].

In this study, *Bacillus* isolate B29 showed the highest protease clear zone compared with other isolates. Protease is a secreted enzyme that can break down peptide bonds in protein nutrients into amino acids for supporting digestibility and absorbability in aquatic animals. Hence, the increase of *Bacillus* isolate B29 in the gut microbiota of fish via feed supplementation could contribute to the digestion of protein, and it might be possible to apply it in aquatic animal feed to enhance feeding efficiency and weight gain of fish. In addition, protease produced by *Bacillus* spp. can also decompose organic matter in rearing water to improve water quality via bioremediation which offers an alternative method for preventing pathogens [46,71,72]. Based on our previous results, *Bacillus* isolate B29 was further selected for antimicrobial susceptibility and pathogenicity tests. The result revealed that *Bacillus* isolate B29 was sensitive to all tested antibiotics. Since dietary probiotic supplementation is considered a possible reservoir of antibiotic-resistant genes for distribution in the gut microbiome of aquatic animals, it leads to the rapid emergence of antibiotic-resistant microbial strains. This information provides basic knowledge concerning the resistance ability of probiotic candidates to antibiotics, and the absence of transferable antibiotic resistance in probiotics assures the safety of their concerns over risks associated with hazardous residues on aquaculture products. Besides, the toxicity and pathogenicity of *Bacillus* isolate B29 were evaluated by determining the lethal dose (LD50) in Nile tilapia (in vivo). During the experiment, no mortality, physical, clinical, or pathological changes were observed in the experimental fish. Therefore, *Bacillus* isolate B29 is considered safe for Nile tilapia and can be applied as feed supplementation for further study. In addition, several studies have reported that *Bacillus* spp., as a growth promoter, could produce some essential nutrients, such as amino acids, vitamins, and digestive enzymes, to support the growth of animals [73,74].

According to biological activity results, the potential probiotic *Bacillus* isolate B29 was selected to construct the recombinant probiotic expressing CC chemokine. Such a recombinant probiotic could provide a beneficial probiotic effect of wild-type *B. subtilis* alongside the specific function of the CC chemokine gene to simultaneously enhance immunity and microbiota modulation of Nile tilapia. This study examined how wild-type and recombinant probiotics affected the innate immune system components of Nile tilapia, which included ACH50, lysozyme, total Ig, and phagocytic activity. Innate immunity is the first line of defense against pathogens, subdivided into three compartments: the physical barriers, the humoral defense response, and the cellular defense response [75,76]. The humoral innate immune response consists of multiple vital molecules, including lytic enzymes, complement proteins, and cytokines [77], whereas the cellular defenses include a variety of immune cells, such as phagocytic cells (neutrophils, macrophages, monocytes, and dendritic cells), nonspecific cytotoxic cells (Natural killer cells; NK cells), and polymorphic nuclear cells or granulocytes (basophils, mast cells) [78]. The present study indicated that the oral administration of either wild-type or recombinant *Bacillus* isolate B29 expressing CC chemokine could enhance total Ig, ACH50, lysozyme, and phagocytic activity, suggesting that *Bacillus* isolate B29 could interact with both humoral and cellular components of innate immunity. After oral administration, probiotics were passed through the GI tract and recognized by Toll-like receptors (TLRs) on intestinal epithelial cells. This phenomenon could trigger immune signaling pathways, including the production of cytokines to improve the systemic immune response in fish. Notably, fish fed a diet containing recombinant *Bacillus* isolate B29 expressing CC chemokine exhibited a prolonged ACH50 response period compared to the wild-type strain. This result may suggest that the upregulation of CC chemokine leads to the release of antibodies and other proteins that activate the complement system and the subsequent increases in phagocytosis by opsonins. In this study, wild-type strain and recombinant probiotic supplementation in diet could increase lysozyme levels after 15 and 30 days of a feeding trial. Like higher vertebrates, fish phagocytes also possess lysozyme that can be bactericidal agents following phagolysosome formation. Lysozyme is a hydrolytic enzyme that can hydrolyze the beta-1,4 glycosidic bonds of the peptidoglycan layer in the bacterial cell wall. The increasing lysozyme activity can indicate the body’s ability to eliminate microorganisms, especially Gram-positive bacteria. However, stress, nutrition, and other factors can also increase this activity [79,80,81,82].

Generally, if innate defenses cannot protect the fish from invading pathogens, the adaptive immune system is the next vital mechanism to eliminate pathogens that enter the fish’s body. In the adaptive immune system, immunoglobulins (Ig) or antibodies produced by B lymphocytes are key elements of the humoral immune system. When the exogenous antigen is engulfed by endocytosis and processed by antigen presenting cells (APCs), the processed antigen epitope is presented on the APC cell surface to activate the specific immune system. Subsequently, cytokines are produced and secreted to activate APCs, mainly B cells, for proliferation and differentiation into the bloodstream [83]. In this study, both wild-type and recombinant *Bacillus* isolate B29 expressing CC chemokine supplementation in diet could increase total Ig levels after 7, 15, and 30 days of feeding. This indicated that dietary probiotic supplementation could enhance immunoglobulin (Ig) production, suggesting that *Bacillus* isolate B29 may impact both innate and adaptive immune responses in Nile tilapia. Interestingly, only fish fed the recombinant probiotic diet exhibited a high total Ig level at 6 h, suggesting that CC chemokine may act as a chemoattractant for immunoglobulin trafficking within the bloodstream. The early up-regulation (at 6 h) of total Ig level revealed the potential pro-inflammatory roles for CC chemokine in response to probiotics.

Phagocytosis is a mechanism directly involved in the innate immune system to engulf pathogens by phagocytes, including neutrophils, monocytes, tissue macrophages, and dendritic cells (DCs). Generally, monocytes are prominently found in the blood and play important roles as the first phagocytes that invading microorganisms encounter. This study demonstrated that dietary supplementation with either wild-type or recombinant *Bacillus* isolate B29 expressing CC chemokine significantly increased the PA of the Nile tilapia phagocytes, suggesting that both wild-type strain and recombinant probiotic are capable of inducing phagocytosis. Notably, dietary supplementation with recombinant *Bacillus* isolate B29 expressing CC chemokine exhibits a significant increase in PI compared to the control and wild-type groups, indicating that CC chemokine can stimulate phagocytes by enhancing phagocyte efficiency to engulf pathogens. This result strongly indicates that fish fed with recombinant *Bacillus* isolate B29 expressing CC chemokine could enhance the PA and PI of Nile tilapia phagocytes more effectively than the wild-type strain and the control diet, respectively. Based on current information, it is clear that chemokine can regulate phagocyte activity, including tissue homeostasis, inflammation, wound healing, and host-pathogen interaction, through the interaction between their receptors and phagocytic cells. This interaction is essential for triggering the rearrangement of actin-containing structures required for cell motility and contributing to phagocyte differentiation, proliferation, and polarization [84,85]. For example, Kono et al. [86] suggested that the production of superoxide anion generated in phagocytic bactericidal activity in kidney leukocytes of Japanese flounders can be enhanced by administering the Japanese flounder CC-chemokine gene (pCMV-CCC), due to the fact that CC chemokine has been recognized as a humoral component that plays functional roles in attracting a variety of phagocytic cells to sites of infection. CC chemokine can play roles in normal physiological conditions (homeostatic or constitutively expressed chemokines) and pathological processes [87,88]. Under normal conditions, CC chemokine plays homeostatic roles in immunosurveillance for cellular maintenance or development in several tissues [31,89]. However, the immunomodulatory features of probiotics are significantly influenced by variables such as their source, type, dose, and duration of administration.

## 5. Conclusions

In this study, we identified and characterized the potential *Bacillus* spp. isolated from Nile tilapia’s intestine. Our finding revealed that *Bacillus* isolate B29, closely related to *B. subtilis*, exhibited several desirable probiotic properties and was selected as a host to produce the heterologous protein. This provides valuable information about the wild-type *B. subtilis* and genetically engineered *B. subtilis* strains designed to express CC chemokine proteins as a feed supplementation to stimulate the immune response in Nile tilapia. The dietary supplementation of either wild-type or recombinant probiotics could enhance immune responses in Nile tilapia. However, recombinant probiotics are well-estimated with a stronger protective effect than the wild-type strain. Thus, the enhancers of desirable heterologous proteins in potential probiotics may be an alternative approach to improve immune responses in Nile tilapia. Although the efficacies of wild-type and recombinant *Bacillus* isolate B29 expressing CC chemokine have been confirmed in this study, further studies are needed to determine whether the effect on disease resistance is significant using the Nile tilapia model in vivo. It could be a novel prophylactic strategy for the intensive aquaculture industry, which has applied drugs and chemicals. In addition, the screening of an optimal signal-peptide is required for further study to produce the efficient secretory of heterologous protein.

## Figures and Tables

**Figure 1 animals-13-00986-f001:**

Inhibitory and/or colonization activities of 1 × 10^6^ and 1 × 10^8^ CFU/mL of *Bacillus* isolate B29 against: (**a**) *A. hydrophila*; (**b**) *S. agalactiae*; (**c**) *S. iniae* at 72 h. The red box indicated inhibitory effects against each pathogenic bacterium. Colonization activity against each pathogenic bacteria was measured in millimeters.

**Figure 2 animals-13-00986-f002:**
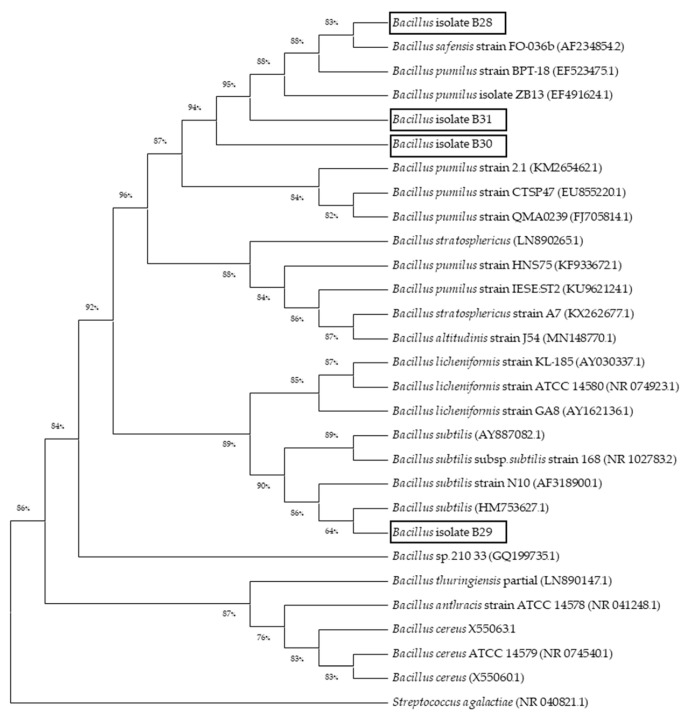
A phylogenetic tree analysis of 16s rRNA gene sequence of four *Bacillus* isolated from Nile tilapia’s intestine and other known *Bacillus* species from GenBank database. The tree was constructed using 1000 bootstrap replications with MEGA X using maximum-likelihood phylogeny method with MUSCLE alignment [35,36]. A discrete Gamma distribution was used to model evolutionary rate differences among sites. The percentages of replicate trees in which the associated taxa clustered together in the bootstrap test (1000 replicates) are shown above the branches. The GenBank accession numbers are provided in brackets.

**Figure 3 animals-13-00986-f003:**
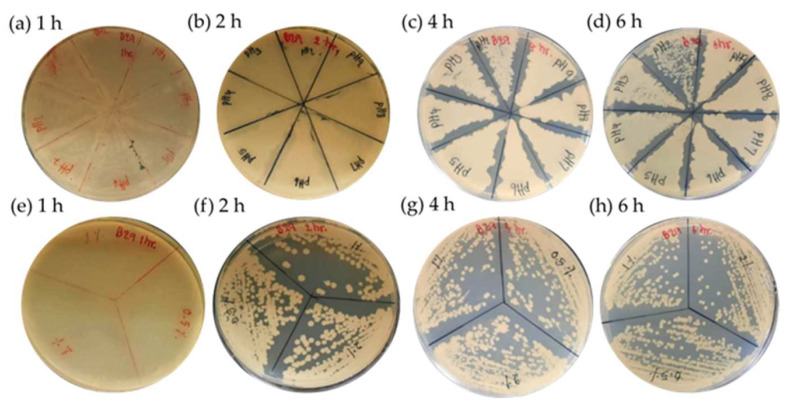
pH and bile salts tolerance test of *Bacillus* isolate B29; (**a**–**d**) pH tolerance test at pH 2–9; (**e**–**h**) bile salts tolerance test at 0.5%, 1%, and 2% for 1, 2, 4, and 6 h, respectively.

**Figure 4 animals-13-00986-f004:**
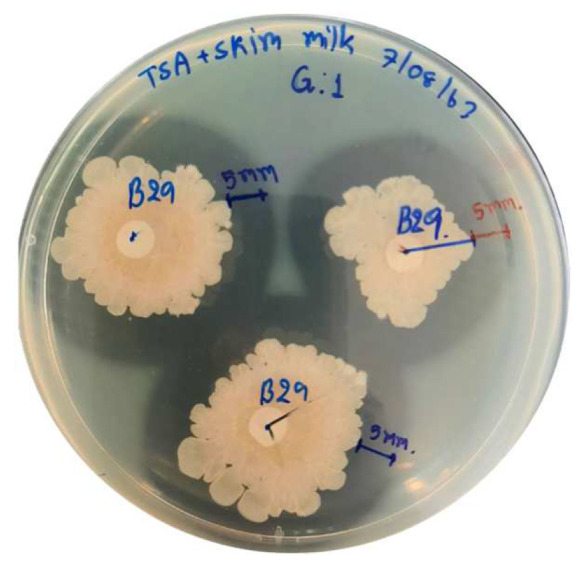
Protease-producing capacity of *Bacillus* isolate B29 on skim milk agar.

**Figure 5 animals-13-00986-f005:**
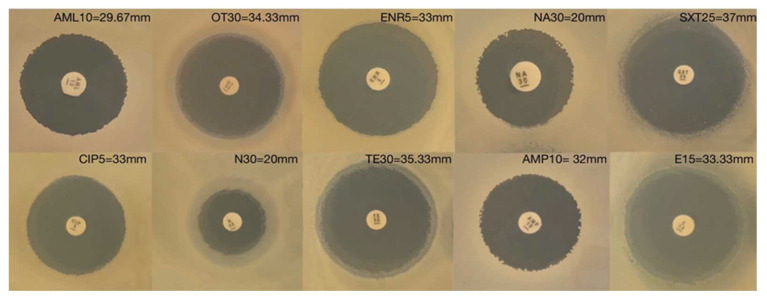
Antibiotic susceptibility of *Bacillus* isolate B29 to commercial antibiotics, including ampicillin (AMP10), neomycin (N30), oxytetracycline (OT30), ciprofloxacin (CIP5), enrofloxacin (ENR5), erythromycin (E15), sulphamethoxazole (SXT25), tetracyclin (TE30), amoxycillin (AML10), and nalidixic acid (NA30).

**Figure 6 animals-13-00986-f006:**
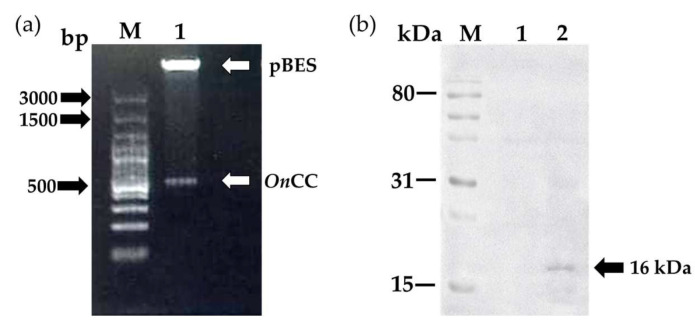
Restriction enzymes (*Bam*HI, *Hind*III) digestion (**a**) Western blot analysis of pBES*On*-CC (**b**); lane M: protein marker, 1: *Bacillus* isolate B29 (Negative control), 2: pBES*On*-CC.

**Figure 7 animals-13-00986-f007:**
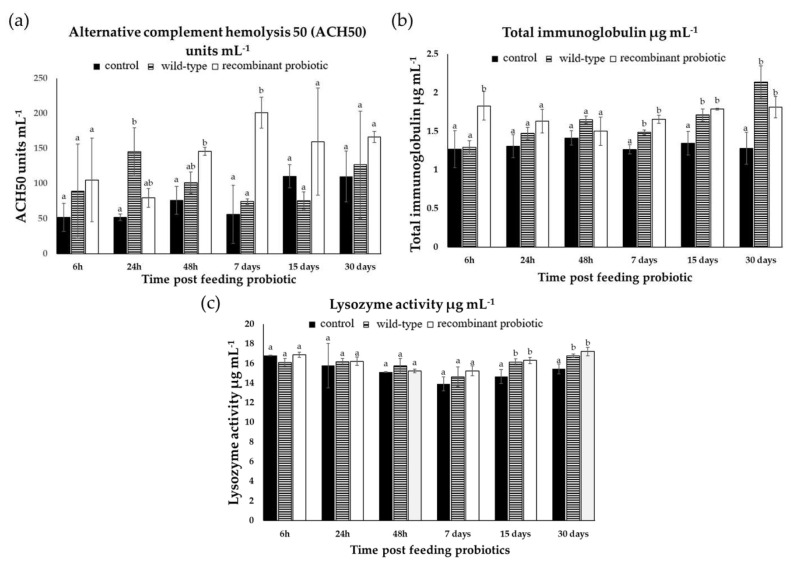
Immune parameters of Nile tilapia fed experimental diets. (**a**) Alternative complement hemolysis 50 (ACH50); (**b**) total immunoglobulin; (**c**) lysozyme activity of Nile tilapia fed with experimental diets for 30 days (n = 5). Bars with different letters indicate significant differences (*p* < 0.05).

**Figure 8 animals-13-00986-f008:**
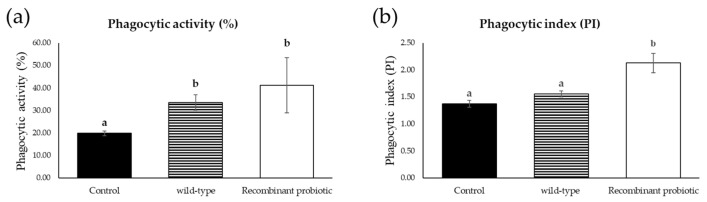
Effect of dietary probiotic supplementation on phagocytic activity and phagocytic index of phagocytic cells in PBLs (n = 5). (**a**) Phagocytic activity (%); (**b**) phagocytic index (PI). Bars with different letters indicate significant differences (*p* < 0.05).

**Table 1 animals-13-00986-t001:** Sequences of primers used in this study.

Primer Name	5′ to 3′ Nucleotide Sequences	Purposes
27F	AGAGTTTGATCMTGGCTCAG	Nucleotide sequencing
1492R	TACGGYTACCTTGTTACGACTT	Nucleotide sequencing
H-B-*On*CCF	AAGCTTGGATCCTCTGGATCAGATGAAAAACT	Cloning
H-X-*On*CCR	AAGCTTCTCGAGCTCGTTCCTGCTTTCTGAAA	Cloning
Beta-actinF	ACAGGATGCAGAAGGAGATCACAG	qRT-PCR
Beta-actinR	GTACTCCTGCTTGCTGATCCACAT	qRT-PCR
*On*CCF	TGGGTTCGTGCCACGATTGTTGCA	qRT-PCR
*On*CCR	TGAAGGAGAGGCGGTGGATGTTAT	qRT-PCR

**Table 2 animals-13-00986-t002:** Probiotic properties of candidate *Bacillus* spp. isolated from the intestine of Nile tilapia.

Isolate	Gram’s Stain	Shape	Catalase Test	Species as Identified by Blast N	Accession No.	Percent Identity (%)
B14	Positive	Bacilli	Positive	*Lysinibacillus fusiformis* strain BN-4	JN039176.1	76.85
B18	Positive	Bacilli	Positive	*Bacillus subtilis* strain MBF4	AB872006.1	81.78
B19	Positive	Bacilli	Positive	*Bacillus pumilus* strain CMF1C	KF640223.1	75.65
B20	Positive	Bacilli	Positive	*Bacillus safensis* strain Bs16	JN699024.1	76.38
B21	Positive	Bacilli	Positive	*Bacillus* sp. ZK10	KJ191434.1	96.08
B22	Positive	Bacilli	Positive	*Bacillus pumilus strain IP5*	KY621522.1	96.66
B23	Positive	Bacilli	Positive	*Bacillus* sp. CRRI-56	JN592473.1	97.24
B24	Positive	Bacilli	Positive	*Bacillus* sp. strain SZ170	KU986705.1	98.65
B27	Positive	Bacilli	Positive	*Bacillus pumilus* strain K26	KU922444.1	98.67
B28	Positive	Bacilli	Positive	*Bacillus pumilus* strain 2.1	EF523475.1	99.87
B29	Positive	Bacilli	Positive	*Bacillus subtilis* strain WSE-KSU303	AY887082.1	99.31
B30	Positive	Bacilli	Positive	*Bacillus pumilus* strain HNS75	EF491624.1	99.87
B31	Positive	Bacilli	Positive	*Bacillus stratosphericus* strain B89	LN890265.1	97.93

**Table 3 animals-13-00986-t003:** Protease-producing capacity of the probiotic candidates.

Isolate	Diameter of Clear Zone (mm)
B28	No clear zone
B29	5.00 ± 0.00
B30	4.67 ± 0.58
B31	4.67 ± 0.58

**Table 4 animals-13-00986-t004:** Expression levels of CC chemokine mRNA in the intestine of Nile tilapia that were fed experimental diets.

Days Post Feeding	Treatment		
	Control	Wild-Type	Recombinant Probiotic
1	0.48 ± 0.03 ^a^	0.45 ± 0.03 ^a^	0.55 ± 0.04 ^b^
2	0.39 ± 0.05 ^a^	0.46 ± 0.06 ^ab^	0.54 ± 0.01 ^b^
7	0.42 ± 0.01 ^a^	0.43 ± 0.03 ^a^	0.49 ± 0.01 ^b^
15	0.42 ± 0.02 ^a^	0.44 ± 0.02 ^ab^	0.47 ± 0.02 ^b^
30	0.48 ± 0.03 ^a^	0.54 ± 0.04 ^ab^	0.59 ± 0.04 ^b^

Means with different superscripts in each row differ significantly from each other (*p* < 0.05). The mRNA level of CC chemokine gene was normalized to the β-actin level after transforming as log 10.

## Data Availability

Not applicable.

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
