# Peer review of "Isolation of Potential Probiotic Bacillus spp. from the Intestine of Nile Tilapia to Construct Recombinant Probiotic Expressing CC Chemokine and Its Effectiveness on Innate Immune Responses in Nile Tilapia"

_animals, 2023, doi:10.3390/ani13060986_

Round 1

Reviewer 1 Report

The current manuscript presents a novel therapeutic alternative in aquaculture by using recombinant probiotics. The context and application of results are well described and with significant impact for the scientific community and fish production sector. The introduction is well written, the methodology is descriptive and detailed. The discussion is in line with the results. Therefore, I recommend the manuscript for publication, pending some points to be taken into consideration.

Minor points to be considered by the authors:

        In line 64 it is mentioned “a member of the genus Bacillus…”. Do the authors mean, “members of the genus Bacillus” or one in specific? If so, the authors could give some examples (B. subtilis, B. amyloliquefaciens), as some of the members of Bacillus can be considered as pathogen to some fish species (e.g, B. cereus https://doi.org/10.1016/j.hal.2020.101771).

-          Lines 113-114: the sentence needs re-arrangement. It reads as the commercial diet is the supplement of Jerusalem artichoke.

-          Line 118: how many Nile tilapia were used to isolate bacterial species?

-          In sections 2.1.5 and 2.1.6 there is no indication if the methods were performed in triplicates as other methods.

-          Line 270: all experimental diets were prepared weekly. I assumed that all the commercial diet was from the same supplier/batch and most importantly, that all the supplementation was made from the same colony of B29 stored in section 2.1.2. Still, can it be emphasized?

-          Small typos: “ad libitum” should be in italic (line 113, 255); “ans” (line 489); “added” (line 323) and “diluted” (line 344) looks like it was used in different font size.

-          Line 298: is the primers information for CC chemokine in reference [40]? It is needed to support that results from section 3.9 were from the host and not from bacteria.

-          Line 442: “best” is a subjective term. It can be elaborated. What defines the results as such?

-          Table 4: what are the units for the expression levels? Use a brief explanation based on ref [40] normalization.

-          Did the authors observe any effect on growth performance? It could be mentioned in the discussion (line 595).

-          For clarification, in some parts it is mentioned as the fish was supplemented with B. subtilis (eg, lines 612, 619, 639, 653,655, 659, 688). It should be mentioned Bacillus isolate B29 (as in lines 564, 574, 593, 614, 642, 679, among others). Please correct other possible cases and make a well distinguished between the results with isolated B29 and possible uses of B. subtilis. Moreover, the authors could elaborate on the benefits of using a probiotic from donors of the same species as the recipient.

Author Response

Dear Reviewer,

We appreciate the opportunity to improve our manuscript and your valuable comments and suggestions. A point-by-point response to your comments is below. We believe that the revisions prompted by these comments have strengthened our manuscript.

-          In line 64 it is mentioned “a member of the genus Bacillus…”. Do the authors mean, “members of the genus Bacillus” or one in specific? If so, the authors could give some examples (B. subtilisB. amyloliquefaciens), as some of the members of Bacillus can be considered as pathogen to some fish species (e.g, B. cereus https://doi.org/10.1016/j.hal.2020.101771).

Response:          Thank you for your suggestion. We have revised this sentence.

Line 64; A member of the genus Bacillus, including B. subtilis, B. licheniformis, B. amyloliquefaciens, B. pumilus, is considered to be among the important probiotics used in aquacultures that are present inside the fish body and the surrounding environment, such as rearing water and sediment [14,15].

-          Lines 113-114: the sentence needs re-arrangement. It reads as the commercial diet is the supplement of Jerusalem artichoke.

Response:          We have revised this sentence.

Line 113; Fish were fed ad libitum twice a day with a 10.0 g kg−1 Jerusalem artichoke (Helianthus tuberosus) supplemented diet for 45 days.

-          Line 118: how many Nile tilapia were used to isolate bacterial species?

Response:          We have added a number of fish used to isolate probiotics, as shown in the revised version of our manuscript.          

Line 118; Bacterial species were isolated from the intestine of ten fish.

-          In sections 2.1.5 and 2.1.6 there is no indication if the methods were performed in triplicates as other methods.

Response:          We have added this information, as shown in the revised version of our manuscript.                       

Line 163; 2.1.5. Determination of bile salts tolerance

To determine the bile salts tolerance, the isolated bacteria were inoculated in TSB enriched with 0.5, 1, and 2% bile salts and incubated at 30 °C for 1, 2, 4, and 6 h, respectively. During 1, 2, 4, and 6 h incubation, the bacterial viability was carried out by streaking on TSA followed by incubation overnight at 30 °C. Each isolated strain was performed in triplicates.

Line 169; 2.1.6. Determination of pH resistance

To determine pH resistance, the isolated bacteria were inoculated in TSB with vary pH 2, 3, 4, 5, 6, 7, 8, and 9 by HCl or NaOH and incubated at 30 °C for 1, 2, 4, and 6 h, respectively. During 1, 2, 4, and 6 h incubation, the bacterial viability was carried out by streaking on TSA followed by incubation overnight at 30 °C. Each isolated strain was performed in triplicates.

-          Line 270: all experimental diets were prepared weekly. I assumed that all the commercial diet was from the same supplier/batch and most importantly, that all the supplementation was made from the same colony of B29 stored in section 2.1.2. Still, can it be emphasized?

Response:         Within 1 month of the experimental period, we used the same bag/supplier/batch of all the commercial diets, and all the supplementation was made from the same colony of B29.

We have added new information in the manuscript to emphasize them. It included:

Line 225; Bacillus isolate B29 from 2.1.2 was used to prepare competent cells according to the method of Xue et al.[39],

Line 262; Prior to this experiment, wild-type Bacillus isolate B29 and recombinant Bacillus isolate B29 from 2.1.2 and 2.2 were aliquoted and stored in glycerol stocks at -80 °C. Each time of experimental diet preparation, an aliquot of glycerol-stock bacteria was streaked onto a freshly prepared TSA plate and incubated at 37 °C for 18-24 h. After that, the starter culture was prepared by inoculating 5 mL TSB with a single colony of each bacterium and incubated at 37 °C for 18-24 h, with a shaking rate of 180 rpm.

-          Small typos: “ad libitum” should be in italic (line 113, 255); “ans” (line 489); “added” (line 323) and “diluted” (line 344) looks like it was used in different font size.

Response:          Thank you for your observation. We have edited the text following your comments.

-          Line 298: is the primers information for CC chemokine in reference [40]? It is needed to support that results from section 3.9 were from the host and not from bacteria.

Response:          We have added new information, as shown in the revised version of our manuscript.

Line 314; The primers used in this experiment were designed following our previous study [31]. We have included this sentence in the manuscript.

-          Line 442: “best” is a subjective term. It can be elaborated. What defines the results as such?

Response:         Thank you for your suggestion. We have revised this sentence.

Line 456; Based on previous results, including antagonistic activity, pH and bile salts tolerance, and protease-producing capacity, only Bacillus isolate B29 was selected to determine the susceptibility test with ten commercial antibiotics. 

-          Table 4: what are the units for the expression levels? Use a brief explanation based on ref [40] normalization.       

Response:          Thank you for your suggestion. We have revised this sentence.

                        Line 498; Means with different superscripts in each row differ significantly from each other (p < 0.05). The mRNA level of CC chemokine gene was normalized to the β-actin level after transforming as log 10.

-          Did the authors observe any effect on growth performance? It could be mentioned in the discussion (line 595).

Response:         Because the experimental fish in this experiment were fed with probiotics supplemented diet for only 1 month, therefore, we did not add the growth performance result in this report. However, we have another experiment that feeds the fish with our probiotics-supplemented diet for 3 months. The effect on growth performance and disease resistance using Nile tilapia model in vivo will be reported in the next manuscript.  

-          For clarification, in some parts it is mentioned as the fish was supplemented with B. subtilis (eg, lines 612, 619, 639, 653,655, 659, 688). It should be mentioned Bacillus isolate B29 (as in lines 564, 574, 593, 614, 642, 679, among others).

Response:         Thank you for your observation. We have edited the text, as shown in the revised version of our manuscript.

-             Please correct other possible cases and make a well distinguished between the results with isolated B29 and possible uses of B. subtilis. Moreover, the authors could elaborate on the benefits of using a probiotic from donors of the same species as the recipient.

Response:             Line 563: Among Probiotic B. subtilis strains, Bacillus isolate B29 in this study, with remarkable antagonistic activity against three important pathogenic bacteria in Nile tilapia, could be a notable candidate Bacillus spp. with applications in the intensive culture system of Nile tilapia farming to reduce drug and chemical application.

Line 618:  Such recombinant probiotic could provide a beneficial probiotic effect of wild-type B. subtilis alongside the specific function of the CC chemokine gene to simultaneously enhance immunity and microbiota modulation of Nile tilapia.

Reviewer 2 Report

The paper by Chatsirin Nakharuthai and colleagues is interesting and well written, mainly described the investigation of the potential probiotic Bacillus spp. from the intestine of Nile tilapia in order to construct a recombinant probiotic for the enhancement of Nile tilapia immune response. The study is significant and presents some valuable findings related to Bacillus spp. While the data presented supports the authors conclusions, the manuscript would be improved if the following concerns were addressed:

1. Whether CC chemokines can be permanently localized in the intestine? If so, the tilapia's long-term immune response would produce the stress response of the fish body, which is not conducive to the normal production and maintenance of healthy body condition of the fish. If not, how long can the innate immune response last? Whether this immune response time plays a role in production and breeding?

2. In Figure 1, which hour is the inhibitory and/or colonization activities shown in the images?  

3. In Table 1, please make Primer name match 5’ to 3’ nucleotide sequences and Purposes. Besides, there are two Table 2 in the paper. Please confirm it carefully.

4. The phylogenetic analysis is weak and needs to be improved (and the phylogenetic tree figure is impossible to follow). Specifically: (i) they need to use a better sequence alignment method than MEGA X. I suggest MAFFT or Muscle; (ii) given the divergent nature of these sequences I think it would be advisable to remove ambiguously aligned sites using TrimAL or GBlocks; (iii) UPGMA has a relatively large systematic error. The authors should repeat the analysis using a more sophisticated maximum likelihood (ML) method with a good model of amino acid substitution (such as LG+gamma).

5. Why were Fish fed Jerusalem artichoke (Helianthus tuberosus)?

Author Response

Dear Reviewer,

We appreciate the opportunity to improve our manuscript and your valuable comments and suggestions. A point-by-point response to your comments is below. We believe that the revisions prompted by these comments have strengthened our manuscript.

  1. Whether CC chemokines can be permanently localized in the intestine? If so, the tilapia's long-term immune response would produce the stress response of the fish body, which is not conducive to the normal production and maintenance of healthy body condition of the fish. If not, how long can the innate immune response last? Whether this immune response time plays a role in production and breeding?

Response:          According to our previous study (Nakharuthai et al., 2016), the mRNA expression level of CC chemokine in the intestine was up-regulated at 6 to 12 h, and the expression was down-regulated and decreased to baseline at 24 after infection with pathogenic bacteria. Because CC chemokine genes are suppressed by feedback inhibition via membrane proteins (cytokine receptors). Thus, they have a short half-life in response to this pathogen, which is a typical characteristic of vertebrate CC chemokine genes. However, a long-term administration of CC chemokine has never been studied in Nile tilapia (in vivo).

In this study, even though the experimental fish were fed recombinant probiotics for one month, neither a physiological nor adverse effect on immune response was observed. Our result showed that a recombinant probiotic-supplemented diet gained several beneficial effects on the fish’s health. However, further study with longer periods of feeding recombinant probiotics is necessary to investigate. In addition, an effective tool for monitoring the adverse effects of prolonged CC chemokine administration will be used to elucidate this hypothesis further.

Reference; Nakharuthai, C.; Areechon, N.; Srisapoome, P. Molecular characterization, functional analysis, and defense mechanisms of two CC chemokines in Nile tilapia (Oreochromis niloticus) in response to severely pathogenic bacteria. Dev. Comp. Immunol. 2016, 59, 207-228.

  1. In Figure 1, which hour is the inhibitory and/or colonization activities shown in the images? 

Response:         We have added the inhibitory and/or colonization hour in the figure legend.

Figure 1. Inhibitory and/or colonization activities of 1 x 106 and 1 x 108 CFU/mL of Bacillus isolate B29 against; (a) A. hydrophila; (b) S. agalactiae; (c) S. iniae at 72 h. The red box indicated inhibitory effects against each pathogenic bacterium. Colonization activity against each pathogenic bacteria was measured in millimetres.

  1. In Table 1, please make Primer name match 5’ to 3’ nucleotide sequences and Purposes. Besides, there are two Table 2 in the paper. Please confirm it carefully.

Response:          We thank the reviewer for the observation. We have modified the table, as shown in the revised version of our manuscript.

  1. The phylogenetic analysis is weak and needs to be improved (and the phylogenetic tree figure is impossible to follow). Specifically: (i) they need to use a better sequence alignment method than MEGA X. I suggest MAFFT or Muscle; (ii) given the divergent nature of these sequences I think it would be advisable to remove ambiguously aligned sites using TrimAL or GBlocks; (iii) UPGMA has a relatively large systematic error. The authors should repeat the analysis using a more sophisticated maximum likelihood (ML) method with a good model of amino acid substitution (such as LG+gamma).

Response:          We have modified a phylogenetic tree following your suggestions, as shown in the revised version of our manuscript.

  1. Why were Fish fed Jerusalem artichoke (Helianthus tuberosus)?

Response:          We have re-arranged the sentence to: “In this experiment, the experimental fish were fed ad libitum twice a day with a 10.0 g kg−1 Jerusalem artichoke (Helianthus tuberosus) supplemented diet for 45 days.”

According to the previous report of Tiengtam et al., 2015, the result showed that supplementation with Jerusalem artichoke (JA) at 10 g kg−1 positively affected Nile tilapia's growth and health. In addition, JA has great potential for use as prebiotics in fish feed due to its ability to promote the proliferation of beneficial bacteria in the gut. Therefore, JA supplementation could increase the opportunity to isolate probiotic bacteria from fish.

Reference; N. Tiengtam, S. Khempaka, P. Paengkoum, S. Boonanuntanasarn∗. Effects of inulin and Jerusalem artichoke (Helianthus tuberosus) as prebiotic ingredients in the diet of juvenile Nile tilapia (Oreochromis niloticus). Anim. Feed Sci. Technol. 207,120-129.
